# Impact of Enhanced Recovery After Surgery (ERAS) protocol versus standard of care on postoperative Acute Kidney Injury (AKI): A meta-analysis

**Whenzhen Shen**[☯]**, Zehao Wu**[☯]**, Yunlu Wang, Yi Sun, Anshi Wu**\*

Department of Anesthesiology, Beijing Chaoyang Hospital, Capital Medical University, Beijing, China

☯ These authors contributed equally to this work.
\* wuanshi88cy@163.com

**Data Availability Statement:** All relevant data are within the manuscript and its Supporting Information files.

## Abstract

### Background

Acute kidney injury (AKI) is a common postoperative complication with an incidence of nearly 15%. Relatively balanced fluid management, flexible use of vasoactive drugs, multimodal analgesia containing non-steroidal anti-inflammatory drugs are fundamental to ERAS protocols. However, these basic tenants may lead to an increased incidence of postoperative AKI.

### Methods

A search was done in the PubMed, Embase, Cochrane Library and reference lists to identify relevant studies from inception until May 2020 to be included in this study. Effects were summarized using pooled risk ratios (RRs), mean differences (MDs) and corresponding 95% confidence intervals (CIs) with random effect model. Heterogeneity assessment, sensitivity analysis, and publication bias were performed.

### Results

A systematic review of nineteen cohort studies covering 17,205 patients, comparing impact of ERAS with conventional care on postoperative AKI was performed. Notably, the ERAS regimen did not increase the incidence of postoperative AKI compared with standard care (RR: 1.21; 95% CI: 0.96 to 1.52; $I^2$ = 53%). Both goal-directed fluid therapy (RR: 1.26; 95% CI: 0.99–1.61; $I^2$ = 55%) and restrictive fluid management (RR: 1.06; 95% CI: 0.57–1.98; $I^2$ = 60%) had no significant effect on the incidence of postoperative AKI. There was no significant statistical difference between different AKI diagnostic criteria (P = 0.43; $I^2$ = 0%). ERAS group had significantly shorter hospital stay (MD: −1.54; 95% CI: −1.91 to −1.17; $I^2$ = 66%). There was no statistical difference in 30-day readmission rate (RR: 0.98; 95% CI: 0.80 to 1.20; $I^2$ = 42%), 30-day reoperation rate (RR: 0.98; 95% CI: 0.71 to 1.34; $I^2$ = 42%) and mortality (RR: 0.81; 95% CI: 0.59 to 1.11; $I^2$ = 0%) between the two groups.

**Funding:** The author(s) received no specific funding for this work.

**Competing interests:** The authors have declared that no competing interests exist.

## Conclusions

This meta-analysis suggests that ERAS protocols do not increase readmission or reoperation rates and mortality while significantly reducing LOS. Most importantly, the ERAS protocol was shown to have no promoting effect on the incidence of postoperative AKI. Even GDFT and restrictive fluid management cannot avoid the occurrence of postoperative AKI, and the ERAS protocol is still worth recommending and its safety is further confirmed.

## Introduction

Acute kidney injury (AKI) is a common complication in hospitalized patients, which has a negative impact on the prognosis of patients. Indeed, AKI is associated with increased length of stay and cost, and its occurrence is an independent risk factor for patient death, which can increase mortality by more than 50%. AKI is also a common postoperative complication after major abdominal surgery, with an incidence of nearly 15% [1, 2]. It is characterized by a dramatic decline in renal function, eventually accompanied by disruption of electrolyte, fluid, and metabolic homeostasis. The severity of AKI ranges from mild changes in biochemical markers to severe renal impairment requiring temporary or permanent renal replacement therapy [3]. Therefore, postoperative AKI is of special significance and can be used as a measurable index of perioperative damage and an important potential intervention target [1, 2, 4].

In 1997, "fast-track surgery" (FTS) was first proposed by Professor Kehlet, University of Copenhagen, Denmark, and in 2005 the European Society for Nutrition and Metabolism (ESPEN) developed a standardized overall perioperative protocol for enhanced recovery after surgery (ERAS) [5]. Essential modalities for ERAS protocol in the perioperative period include elements such as minimization of narcotics with multimodal analgesia, maintenance of euvolemia, early diet resumption, and early ambulation [6]. These programs address patient recovery preoperatively, intraoperatively, and postoperatively with a variety of interventions. The adoption of enhanced recovery after surgery (ERAS) has increased over the past few years, and a number of observational studies have pointed to its safety and ability to reduce hospital stay, overall mortality, and hospital costs [7]. However, as more hospitals have adopted ERAS protocol, controversies surrounding elements of ERAS and specific complication risks have emerged. Relatively balanced fluid management, flexible use of vasoactive drugs, multimodal analgesia containing non-steroidal anti-inflammatory drugs raise concerns that the use of ERAS may lead to an increased incidence of postoperative AKI [8]. Proven by studies that ERAS protocol exacerbates the risk of AKI development after colorectal surgery [9–13]. However, some scholars have objected that ERAS will not aggravate the occurrence of postoperative AKI [14–17]. In order to investigate and address this concern, we used this systematic review of the literature with meta-analysis was to assess the impact of ERAS protocols vs. standard of care on kidney function and the incidence of postoperative AKI.

## Methods

### Search strategy

This systematic review was conducted in accordance with the Preferred Reporting Items for Systematic Reviews and Meta-analyses (PRISMA) statement [18, 19]. The review protocol was registered in PROSPERO (www.crd.york.ac.uk/PROSPERO), and registration number CRD42020187720.

We searched PubMed, Embase and Cochrane Library to identify cohort studies, interventional trials, and reviews that to evaluate the impact of ERAS protocols vs. standard care on postoperative AKI using a broad search strategy. PubMed was searched using the following query: ((enhanced recovery after surgery [Title/Abstract]) OR (ERAS [Title/Abstract]) OR (fast-track surgery [Title/Abstract]) OR (FTS [Title/Abstract])) AND ((acute kidney injury) OR (AKI) OR (acute renal failure) OR (ARF) OR (complications) OR (outcomes)). References of included studies were then scanned to identify additional relevant trials (S1 Appendix).

## Inclusion criteria

The following inclusion criteria shall be met for the included studies: (1) cohort studies; (2) Newcastle-Ottawa Scale ≥7; (3) studies establishing the impact of ERAS protocols versus standard of care on postoperative complications; (4) the presence or absence of postoperative AKI was reported in the study.

Any of the following studies were excluded: (1) studies of emergency surgeries; (2) studies that focused only on a single aspect of the enhanced recovery pathway; (3) case reports, meta-analyses, reviews, protocol studies, or letters; (4) studies that did not report primary outcomes; and (5) data that were insufficient for interpretation by meta-analysis.

## Data extraction

Two reviewers (WS and ZW) independently assessed whether the included studies met the criteria separately. Any dispute was resolved by consulting a third reviewer (YW). The PRISMA flow diagram was used to summarize the study inclusion and selection process.

Extracted data included first author name; the year of publication; country where the study was conducted; study design; subjects' demographic characteristics; number of exposure and control groups included in the study; procedure types; length of stay of exposure and control group and incidence of postoperative AKI in the two groups; intraoperative and postoperative fluid management methods; incidence of 30-day readmission, reoperation and mortality; AKI diagnostic criteria. Two reviewers (ZW and YS) extracted all of the above data, while lead reviewer (AW) checked the extracted data.

## Assessment of risk of bias and study quality

Two authors (WS and YS) were independently responsible for assessments of bias. Funnel plots were performed for all outcome measures to evaluate for possible publication bias.

Two reviewers (ZW and YW) used the Newcastle-Ottawa Scale (NOS) criteria [20] to assess the quality of included cohort studies. Score ranged from 0 to 9 stars. High-quality studies were those with scores of more than 7 stars while the low-quality studies were scores of less than 3 stars and the moderate-quality were 4 to 6 stars.

## Statistical analysis

Cohort study dichotomous outcome variables (e.g., incidence of postoperative AKI, readmission and reoperation rates, and mortality) were pooled using relative risk (RR) and 95% confidence interval (95% CI). Required data will be calculated as necessary from the data or figures presented in the study. For continuous variables (such as: LOS), since consistent or similar measurement means of each original study outcome will not cause great difference in outcome variables, mean difference (MD) and 95% CI are used to combine effect size. The overall effect was assessed by $Z$ test using a random effects model (Inverse Variance method) and statistical significance was determined when the 95% CIs did not include the value of 1.0 for the RR or 0 for the MD [21].

Results of the included cohort studies were summarized qualitatively. Patient data on interventions and postoperative complications were extracted from all included studies. MD, RR and 95% CI were calculated to confirm the association of both regimens with the occurrence of AKI. The Q and $I^2$ statistics were used to determine heterogeneity, $I^2$ was defined as: $100\% \times (Q–df)/Q$, where Q was Cochrane's heterogeneity statistic, df was the degree of freedom, $I^2 < 50\%$ indicated that there was no significant heterogeneity, a fixed-effect model was used, and conversely $> 50\%$ used a random-effect model [22]. Subgroup analyses were performed on the fluid management mode, as well as AKI diagnostic criteria. Sensitivity analyses were performed to test the reliability of the results by removing each study individually and changing effects model of the statistical method (fixed-effect model [Mantel-Haenszel method] vs. random-effect model [Inverse Variance method]). It has previously been shown that the probability of reporting positive results is significantly higher in some studies of specific surgical types such as cardiac surgery than in other types of surgery. Therefore, the authors planned to conduct a "special" sensitivity analysis by removing cardiac surgery studies to confirm the robustness of the pooled analysis results. Publication bias was assessed using funnel plots with simultaneous Egger regression test [23] and Begg adjusted rank correlation test [24]. Because most of the included studies were retrospective cohort studies, statistical power for the primary outcome was calculated using G power software, and the type of power analysis selected was Post hoc: Compute power-achieved given α, sample size, and effect size. Sensitivity analyses, trim and fill analysis were performed by Stata 15 (StataCorp, College Station, TX) to determine whether the results were robust, and data summary analysis and forest plots were using Review Manager 5.3. (The Cochrane Collaboration, UK).

## Results

### Characteristics of included studies

Our initial search yielded 7,441 potentially relevant cohort studies on postoperative complications including AKI occurring with ERAS versus conventional therapy in patients who underwent surgery. After screening and reviewing, 19 studies [9–17, 25–34] met our inclusion criteria. Of the nineteen, seventeen were retrospective and two were prospective cohort studies. Fig 1 shows the flow diagram of article selection at different stages of the systematic review.

The studies had 17,205 subjects that were pooled for meta-analyses, namely 7,766 (45%) participants to some ERAS protocol and 9,439 (55%) controls receiving standard of care. AKI was demonstrated in 831 (461 ERAS protocol and 370 standard treatment) of all subjects after surgery, with an overall incidence of approximately 4.83%. Main patient, outcomes, country and surgical characteristics are reported in Table 1. In most studies, baseline characteristics did not differ significantly between ERAS participants and controls, although wide between study heterogeneity was evident for most of these characteristics. Most patients were treated with open surgery. All studies were run at academic or tertiary referral centers.

Of the included 19 studies, all of studies showed the rate of AKI after surgery, six of these studies [11–13, 27, 30, 32] also reported postoperative complications other than AKI, only 3 studies [16, 33, 34] were not reported a difference in length of stay (LOS) in the ERAS vs. standard group. Rate of 30-day readmission or reoperation reported in 13 studies [9–12, 15, 17, 25–30, 32] and mortality in 8 studies [11, 12, 14, 15, 25, 26, 30, 32]. 11 studies adopted the fluid management mode of goal-directed fluid therapy (GDFT) [10–17, 27, 29, 30], 6 studies used restrictive fluid management intraoperatively [9, 25, 26, 28, 33, 34], and the remaining 2 studies [31, 32] did not elaborate the fluid management mode. Ten studies [9, 10, 12–15, 29, 31, 33, 34] adopted the KDIGO criteria to evaluate AKI, which 1.5 times the preoperative baseline value within 30 days after operation based on the latest clinical guidelines from the Kidney Disease: Improving Global

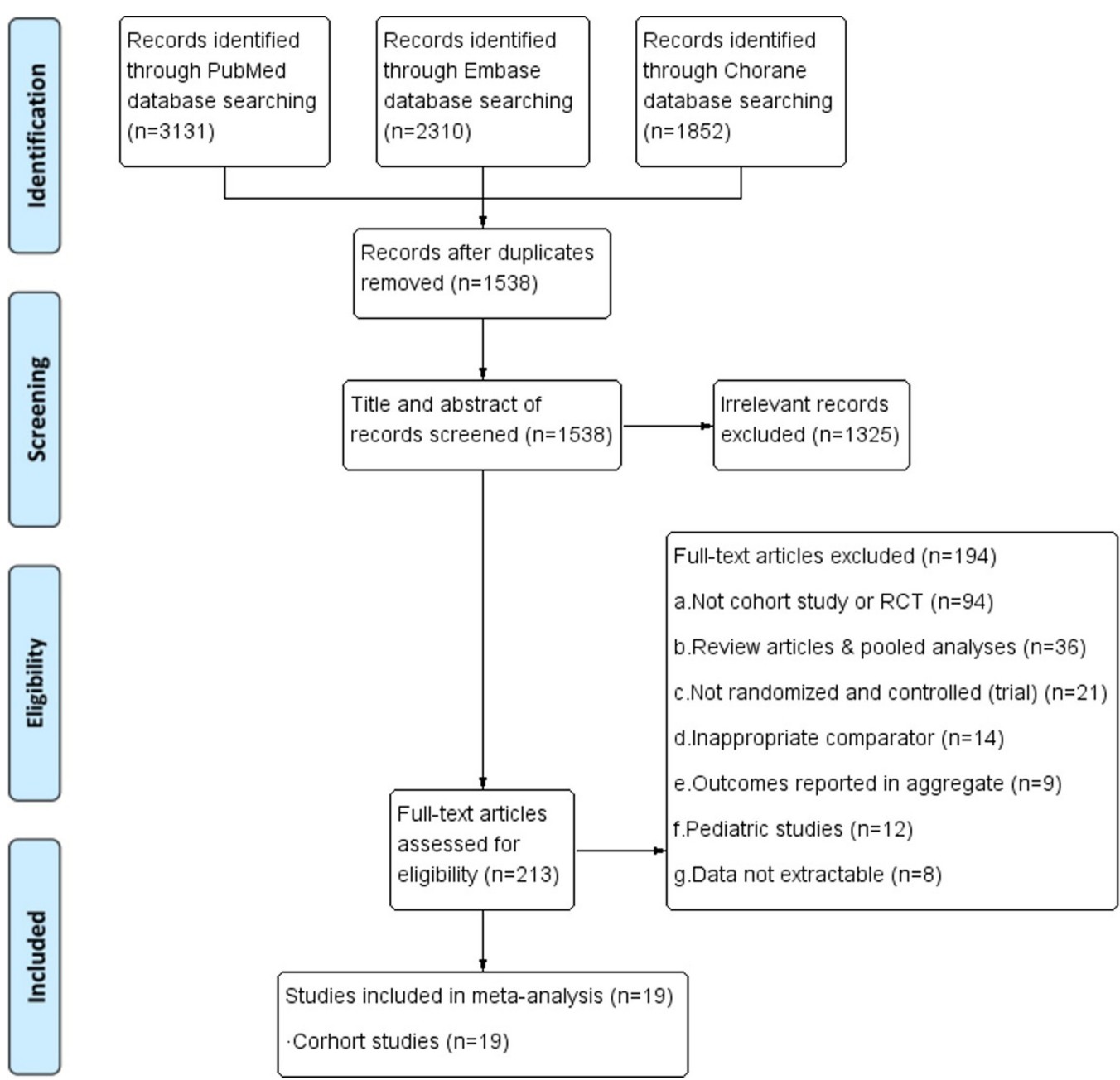

**Fig 1. Preferred reporting Items for systematic reviews and meta-analyses (PRISMA) flow diagram of the process of article selection.** RCT, randomized controlled trial.

Outcomes (KDIGO) criteria [35]. Four studies [11, 16, 17, 30] adopted RIFLE (Risk, Injury, Failure, Loss, and End-stage renal disease classification) [36], EPCO (European Perioperative Clinical Outcome definitions) [37], and NSQIP (National Surgical Quality Improvement Program) [38] criteria to define the occurrence of AKI respectively and the remaining 5 studies [25–28, 32] did not mention the evaluation criteria for AKI. Comparatively speaking, KDIGO criteria are more liberal than other criteria. No major cardiovascular complications such as nonfatal myocardial infarction, stroke or cardiac arrest were reported in all studies.

**Table 1. Details of studies included in this analysis.**

| Authors | Study design | Country | Surgery type | Fluid management mode | Outcome(s) | AKI diagnostic criteria | N. of patients (ERAS/ Standard) | Age (year) (ERAS/ Standard) | BMI (kg/ m² ) (ERAS/ Standard) |
|---|---|---|---|---|---|---|---|---|---|
| Arumainayagam 2008 [25] | retrospective cohort study | UK | Radical cystectomy | Restrictive fluid administration | Rate of AKI | Not mentioned | 56/56 | 65.9/65.9 | - |
| | | | | | LOS | | | | |
| | | | | | Rate of 30-day readmission | | | | |
| | | | | | Rate of 30-day reoperation | | | | |
| | | | | | Mortality | | | | |
| Baldini 2018 [26] | retrospective cohort study | France | Radical cystectomy | Restrictive fluid administration | Rate of AKI | Not mentioned | 41/56 | 67/70 | 26/25 |
| | | | | | LOS | | | | |
| | | | | | Rate of 30-day readmission | | | | |
| | | | | | Mortality | | | | |
| Doyle 2019 [14] | retrospective cohort study | UK | Laparotomy | GDFT | Rate of AKI | KDIGO | 426/303 | 65.8/65.6 | - |
| | | | | | LOS | | | | |
| | | | | | Mortality | | | | |
| Drakeford 2018 [34] | retrospective cohort study | Singapore | Colorectal surgery | Restrictive fluid administration | Rate of AKI | KDIGO | 104/112 | - | - |
| Hassinger 2018 [15] | retrospective cohort study | USA | Colorectal surgery | GDFT | Rate of AKI | KDIGO | 439/461 | 58.17/57.26 | 28.21/27.99 |
| | | | | | LOS | | | | |
| | | | | | Rate of 30-day readmission | | | | |
| | | | | | Mortality | | | | |
| Hawkins 2019 [27] | retrospective cohort study | USA | Colorectal surgery | GDFT | Rate of AKI | Not mentioned | 550/632 | 55/54 | 26.4/27.0 |
| | | | | | Rate of other complications | | | | |
| | | | | | LOS | | | | |
| | | | | | Rate of 30-day readmission | | | | |
| | | | | | Rate of 30-day reoperation | | | | |
| Horres 2017 [16] | retrospective cohort study | USA | Colorectal surgery | GDFT | Rate of AKI | RIFLE | 590/464 | 60/60 | - |
| Koerner 2019 [9] | retrospective cohort study | USA | Colorectal surgery | Restrictive fluid administration | Rate of AKI | KDIGO | 113/196 | 55.93/53 | 28.3/27.47 |
| | | | | | LOS | | | | |
| | | | | | Rate of 30-day readmission | | | | |
| Lu 2020 [29] | retrospective cohort study | USA | Cytoreductive surgery and HIPEC | GDFT | Rate of AKI | KDIGO | 20/11 | 50/47 | - |
| | | | | | LOS | | | | |
| | | | | | Rate of 30-day readmission | | | | |
| Mannaerts 2016 [28] | retrospective cohort study | USA | Bariatric surgery | Restrictive fluid administration | Rate of AKI | Not mentioned | 1313/654 | 42.9/44.2 | 44.3/45.6 |
| | | | | | LOS | | | | |
| | | | | | Rate of 30-day readmission | | | | |
| | | | | | Rate of 30-day reoperation | | | | |

(*Continued*)

**Table 1.** (Continued)

| Authors | Study design | Country | Surgery type | Fluid management mode | Outcome(s) | AKI diagnostic criteria | N. of patients (ERAS/ Standard) | Age (year) (ERAS/ Standard) | BMI (kg/ m$^2$) (ERAS/ Standard) |
|---------|-------------|---------|--------------|----------------------|------------|------------------------|--------------------------------|----------------------------|----------------------------------|
| Marcotte 2018 [10] | retrospective cohort study | USA | Colorectal surgery | GDFT | Rate of AKI | KDIGO | 132/132 | 61.81/61.82 | 30.36/28.64 |
| | | | | | LOS | | | | |
| | | | | | Rate of 30-day readmission | | | | |
| | | | | | Rate of 30-day reoperation | | | | |
| Ripolles-Melchor 2019 [11] | prospective cohort study | Spain | Colorectal surgery | GDFT | Rate of AKI | EPCO | 1304/780 | 68/69 | 26.67/26.76 |
| | | | | | Rate of other complications | | | | |
| | | | | | LOS | | | | |
| | | | | | Rate of 30-day readmission | | | | |
| | | | | | Rate of 30-day reoperation | | | | |
| | | | | | Mortality | | | | |
| Ripolles-Melchor 2020 [30] | prospective cohort study | Spain | Total hip and knee arthroplasty | GDFT | Rate of AKI | EPCO | 1592/4554 | 70/71 | 29.3/29.4 |
| | | | | | Rate of other complications | | | | |
| | | | | | LOS | | | | |
| | | | | | Rate of 30-day readmission | | | | |
| | | | | | Rate of 30-day reoperation | | | | |
| | | | | | Mortality | | | | |
| Salti 2019 [31] | retrospective cohort study | USA | Cytoreductive surgery and HIPEC | Not mentioned | Rate of AKI | KDIGO | 51/51 | - | - |
| | | | | | LOS | | | | |
| Shim 2020 [12] | retrospective cohort study | Korea | Colorectal surgery | GDFT | Rate of AKI | KDIGO | 210/210 | 63/65 | 24.2/23.4 |
| | | | | | Rate of other complications | | | | |
| | | | | | LOS | | | | |
| | | | | | Rate of 30-day reoperation | | | | |
| | | | | | Mortality | | | | |
| Sutcliffe 2015 [32] | retrospective cohort study | UK | Pancreaticoduodenectomy | Not mentioned | Rate of AKI | Not mentioned | 65/65 | 67/66 | 27.3/25.4 |
| | | | | | Rate of other complications | | | | |
| | | | | | LOS | | | | |
| | | | | | Rate of 30-day readmission | | | | |
| | | | | | Mortality | | | | |
| Varelmann 2019 [33] | retrospective cohort study | USA | Cardiac surgery | Restrictive fluid administration | Rate of AKI | KDIGO | 107/173 | - | - |
| Webb 2020 [17] | retrospective cohort study | USA | Cytoreductive surgery and HIPEC | GDFT | Rate of AKI | NSQIP | 81/49 | 54.4/56.0 | - |
| | | | | | LOS | | | | |
| | | | | | Rate of 30-day readmission | | | | |
| | | | | | Rate of 30-day reoperation | | | | |

(*Continued*)

**Table 1.** (Continued)

| Authors | Study design | Country | Surgery type | Fluid management mode | Outcome(s) | AKI diagnostic criteria | N. of patients (ERAS/ Standard) | Age (year) (ERAS/ Standard) | BMI (kg/ m² ) (ERAS/ Standard) |
|---------|--------------|---------|--------------|-----------------------|------------|-------------------------|----------------------------------|------------------------------|---------------------------------|
| Wiener 2020 [13] | retrospective cohort study | USA | Colorectal surgery | GDFT | Rate of AKI | KDIGO | 572/480 | 57.1/57.1 | 27.7/26.2 |
| | | | | | Rate of other complications | | | | |
| | | | | | LOS | | | | |

Abbreviations: GDFT, goal-directed fluid therapy; LOS, length of stay; AKI, acute kidney injury; KDIGO, the Kidney Disease: Improving Global Outcomes criteria; RIFLE, Risk, Injury, Failure, Loss, and End-stage renal disease classification; EPCO, European Perioperative Clinical Outcome definitions; NSQIP, National Surgical Quality Improvement Program; BMI, body mass index.

### Risk of bias within studies

Cohort studies were evaluated for bias based on the Newcastle-Ottawa Scale (Fig 2). Only 4 studies achieved the maximum of 9 stars, the remaining 15 studies achieved 7 stars, and none of studies assessed as moderate to low quality trial. The majority of bias was found in the comparability of cohorts. Most studies showed that there were no statistically significant differences between the cohorts.

### The incidence of postoperative acute kidney injury

All nineteen studies including 17205 participants assessed the impact of enhanced recovery after surgery protocols versus standard of care on postoperative AKI. As shown in Fig 3, ERAS protocol was not associated with an increased incidence of postoperative AKI compared with standard protocol (RR: 1.21; 95% CI: 0.96 to 1.52; $I^2$ = 53%).

Funnel plot or Egger regression test (P = 0.496) did not show any publication bias for primary outcome (the rate of AKI). The trim and fill analysis did not suggest any signs of asymmetry. Although the results of primary outcome indicated statistically moderately heterogeneity, sensitivity analysis of the rate of AKI by individually removing specific studies that could affect the outcome and changing the effect model did not change the results and the sensitivity analysis indicated that the results of primary outcome were robust (S1 Fig). Because the sample size of this study is sufficient, the results suggest that the statistical power is high (power = 0.9999764), so the possibility that the significant results reported in this study truly reflect the real effect is high.

Subgroup analyses were carried out to evaluate the factors that affected heterogeneity.

Fluid management. The subgroup analysis of incidence of postoperative AKI, including 17205 participants from all studies, was stratified by different modes of fluid management, and as can be seen from the results, among 11 studies that performed the ERAS protocol for GDFT, the incidence of postoperative AKI was not increased and there was not statistically significant difference in the ERAS protocol compared with the standard protocol and close to the pooled results (RR: 1.26; 95% CI: 0.99 to 1.61; $I^2$ = 55%); similarly, the incidence of postoperative AKI did not differ significantly between the ERAS protocol and the standard care in 6 studies that performed restrictive fluid management (RR: 1.06; 95% CI: 0.57 to 1.98; $I^2$ = 60%) and the remaining 2 studies that did not mention fluid management modalities (RR: 0.64; 95% CI: 0.19 to 2.11; $I^2$ = 11%). Since there was no heterogeneity among the three subgroups, fluid management mode was not a source of heterogeneity in the primary outcome, and different fluid management modes did not affect the incidence of postoperative AKI with the ERAS protocol (Fig 3).

| | Representativeness of exposed cohort | Selection of non exposed cohort | Ascertainment of exposure | Demonstration that outcome was not present at start of study | Comparability of cohorts | Assessment of outcome | Adequate time of follow-up | Complete follow-up of cohort | Total score |
|---|---|---|---|---|---|---|---|---|---|
| Arumainayagam 2008 | + | + | + | + | – | – | + | + | + | 7 |
| Baldini 2018 | + | + | + | + | – | – | + | + | + | 7 |
| Doyle 2019 | + | + | + | + | – | – | + | + | + | 7 |
| Drakeford 2018 | + | + | + | + | – | – | + | + | + | 7 |
| Hassinger 2018 | + | + | + | + | – | – | + | + | + | 7 |
| Hawkins 2018 | + | + | + | + | + | + | + | + | + | 9 |
| Horres 2017 | + | + | + | + | – | – | + | + | + | 7 |
| Koerner 2018 | + | + | + | + | + | + | + | + | + | 9 |
| Lu 2019 | + | + | + | + | – | – | + | + | + | 7 |
| Mannaerts 2015 | + | + | + | + | – | – | + | + | + | 7 |
| Marcotte 2018 | + | + | + | + | + | + | + | + | + | 9 |
| Ripolles-Melchor 2019 | + | + | + | + | – | – | + | + | + | 7 |
| Ripolles-Melchor 2020 | + | + | + | + | – | – | + | + | + | 7 |
| Salti 2019 | – | + | + | + | – | + | + | + | + | 7 |
| Shim 2020 | + | + | + | + | + | + | + | + | + | 9 |
| Sutcliffe 2015 | + | + | + | + | – | – | + | + | + | 7 |
| Varelmann 2019 | + | + | + | + | – | – | + | + | + | 7 |
| Webb 2019 | + | + | + | + | – | – | + | + | + | 7 |
| Wiener 2020 | + | + | + | – | + | – | + | + | + | 7 |

**Fig 2. Assessment of bias in cohort studies.** + denotes low risk of bias,–denotes high risk of bias.

Diagnostic criteria. A total of 17,205 subjects from all 19 studies were analyzed according to different diagnostic criteria for AKI. Ten studies used KDIGO criteria to evaluate the occurrence or absence of AKI. The analysis results showed that in the studies of KDIGO evaluation criteria, there was no statistical difference in the incidence rate of postoperative AKI between ERAS protocol and standard protocol (RR: 1.36; 95% CI: 0.99 to 1.86; $I^2$ = 66%); Consistently, 4 studies using non-KDIGO criteria (RR: 0.97; 95% CI: 0.64 to 1.47; $I^2$ = 46%) and 5 studies not mentioning the criteria (RR: 1.15; 95% CI: 0.65 to 2.04; $I^2$ = 12%) all demonstrated that there was no significant difference in the incidence rate of postoperative AKI between ERAS protocol and standard protocol. From the above results, we can conclude that the different

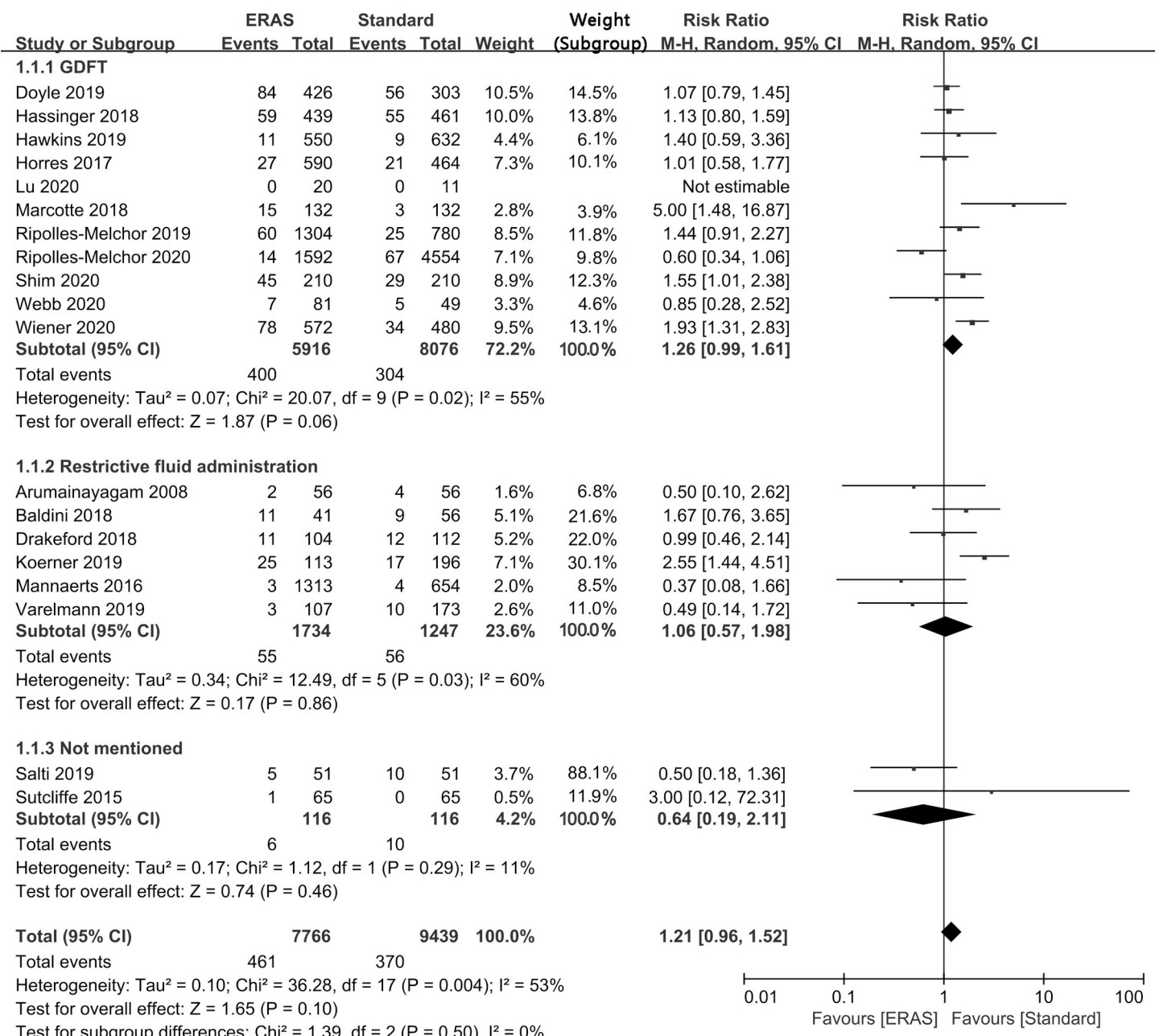

**Fig 3. Results of subgroup analysis of the effect of different fluid management modes on the incidence of postoperative AKI in an ERAS protocol.** CI = confidence interval; ERAS = enhanced recovery after surgery; GDFT = goal-directed fluid therapy; RR = risk ratio.

diagnostic criteria for AKI did not affect the incidence of postoperative AKI, and there was no statistically significant difference in the incidence of postoperative AKI between the ERAS protocol and the standard protocol (Fig 4).

## Length of stay (LOS)

A total of 15,655 subjects treated with either ERAS or standard care in 16 studies were included for the analysis of LOS. An MD of −1.54 (95% CI: −1.91 to −1.17; I² = 66%) was obtained from

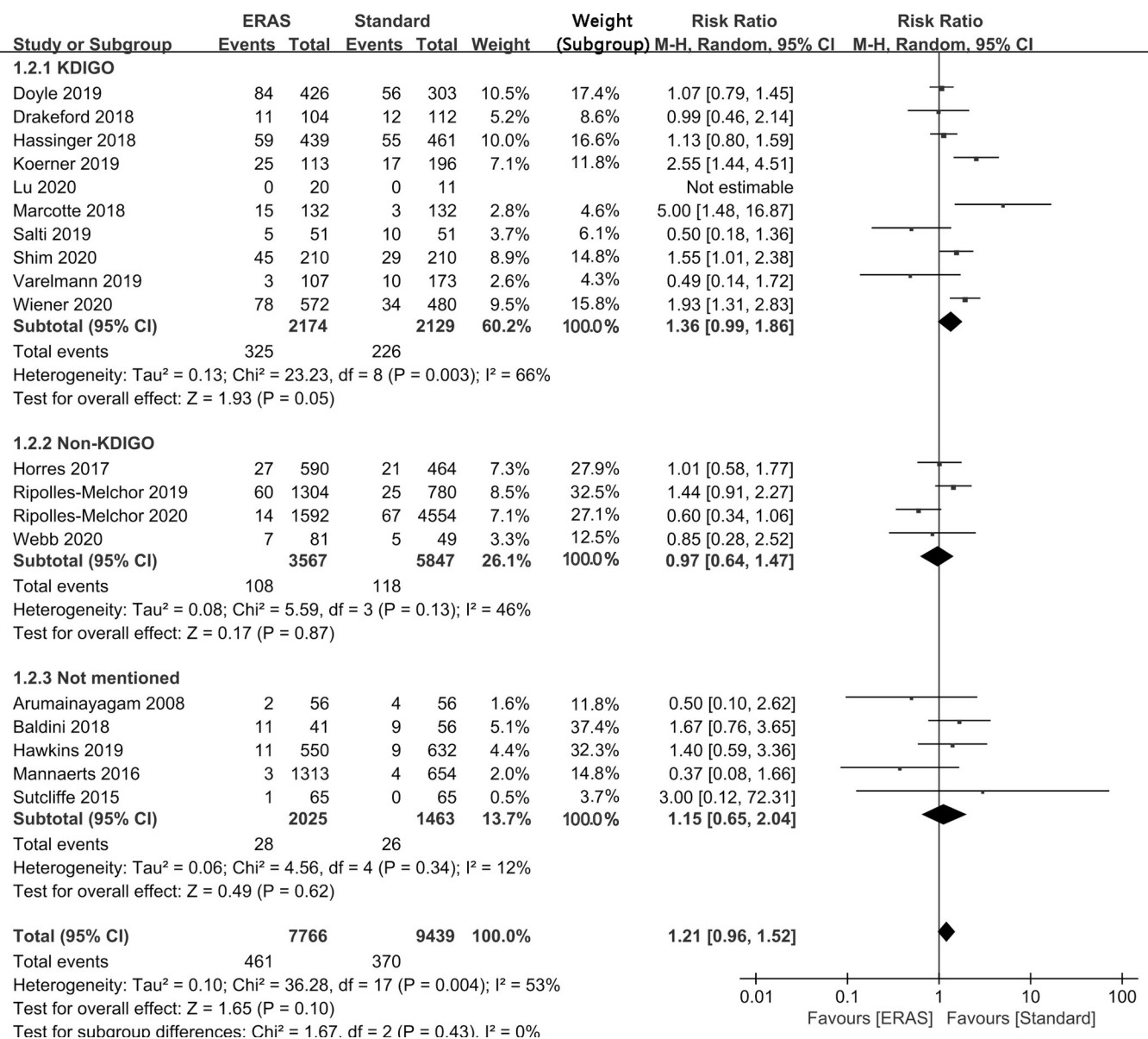

**Fig 4. Results of subgroup analysis results of the effect of ERAS protocol on the incidence of postoperative AKI by different diagnostic criteria for AKI.**
CI = confidence interval; ERAS = enhanced recovery after surgery; KDIGO = the Kidney Disease: Improving Global Outcomes criteria; RR = risk ratio.

analysis of LOS of patients under ERAS and conventional care (Fig 5). The results showed that ERAS group had a shorter hospital stay than conventional care groups (P< 0.001). Although the results indicated statistically significant heterogeneity, the sensitivity analysis showed that the results of hospital stay were robust but the funnel plot and egger regression test suggested publication bias probably (P = 0.01) (S2 Fig). We further tested the publication bias for the studies which included LOS outcome used the Begg adjusted rank correlation, and the results showed that there was no significant publication bias (P = 0.163). In addition, the pooled effect size did not change significantly after using the trim and fill analysis, which was also statistically significant, so we can consider that there is no significant publication bias between the studies included in the LOS outcome measure.

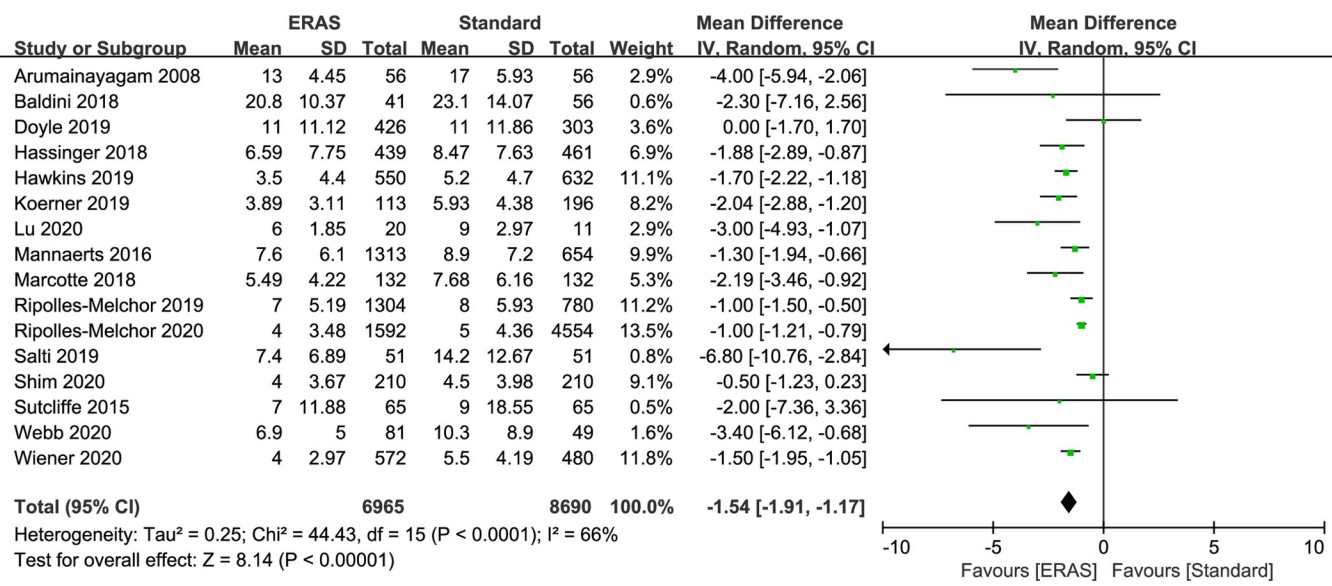

**Fig 5. Results of subgroup analysis results of length of stay in ERAS protocol versus standard care.** CI = confidence interval; ERAS = enhanced recovery after surgery; LOS = length of stay; MD = mean difference.

For this result, we made the following analysis: egger regression test showed P < 0.05, indicating that the funnel plot was asymmetric, but it did not represent the presence of publication bias, that is to say, other reasons caused the asymmetry of the funnel plot. However, as we know, there are many causes of funnel plot asymmetry, which are not caused only by publication bias, such as low-quality small-sample studies, real heterogeneity and artifacts, which can cause funnel plot asymmetry. Although the results of the sensitivity analysis were robust, the heterogeneity of this outcome variable was high, so we considered heterogeneity as a source of funnel plot asymmetry and egger regression test results P < 0.05.

## Readmission rates

Twelve studies reported 30-day readmission rates (13,269 patients). A total of 804 patients were readmitted (391 in ERAS group and 413 in standard care group). When combined, there was no significant difference in readmission rates between ERAS and standard care (RR: 0.98; 95% CI: 0.80 to 1.20; P = 0.84), with slight heterogeneity observed in the studies (P = 0.06; $I^2$ = 42%) (Fig 6). However, the sensitivity analysis suggested that the results of readmission rates were robust and funnel plot or egger regression test did not show any publication bias (P = 0.579) (S3 Fig).

## Reoperation rates

Eight studies reported 30-day reoperation rates (12,305 patients). A total of 417 patients were readmitted (224 in ERAS group and 193 in standard care group). After combining the results, there was no significant difference in reoperation rates between ERAS and standard protocol (RR: 0.98; 95% CI: 0.71 to 1.34; P = 0.89), with slight heterogeneity observed in the studies (P = 0.10; $I^2$ = 42%) (Fig 7). Funnel plot or egger regression test did not show any publication bias (P = 0.985). Trim and fill analysis did not reveal any asymmetry. Sensitivity analyses of reoperation rates by removing each study separately and by changing the effect model did not change the above results (S4 Fig).

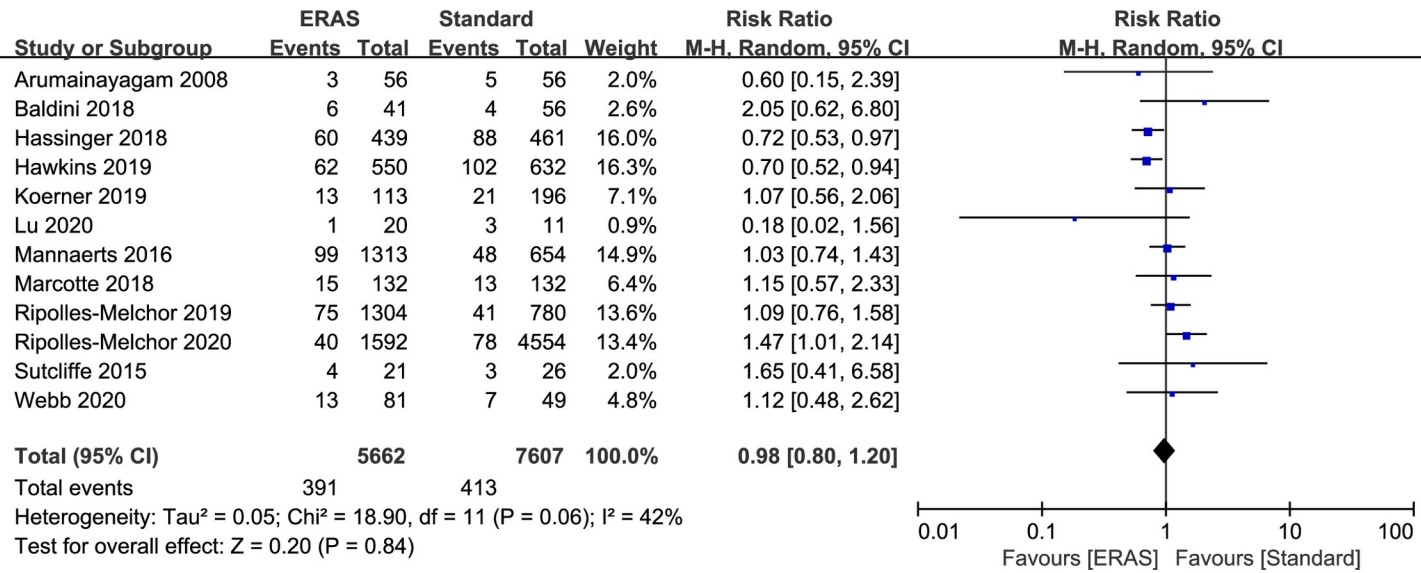

**Fig 6. Results of subgroup analysis results of 30-day readmission rate in ERAS protocol versus standard care.** CI = confidence interval; ERAS = enhanced recovery after surgery; RR = risk ratio.

## Mortality rates

8 studies provided data on mortality rates (10,615 patients). A total of 143 deaths were reported in the studies (75 in ERAS and 68 in standard care group). After combining the results, there was no significant difference in mortality rates between ERAS group and standard care (RR: 0.81; 95% CI: 0.59 to 1.11; P = 0.20) (Fig 8), and no significant heterogeneity observed among the studies (P = 0.97; I² = 0%). The fixed effects model was chosen because there was no heterogeneity. Funnel plot or egger regression test did not suggest any publication bias (P = 0.41). Trim and fill analysis did not show any signs of asymmetry (S5 Fig).

## Discussion

In recent years, more and more medical institutions have implemented ERAS into their management. The ERAS protocol is a standardized perioperative care pathway designed to

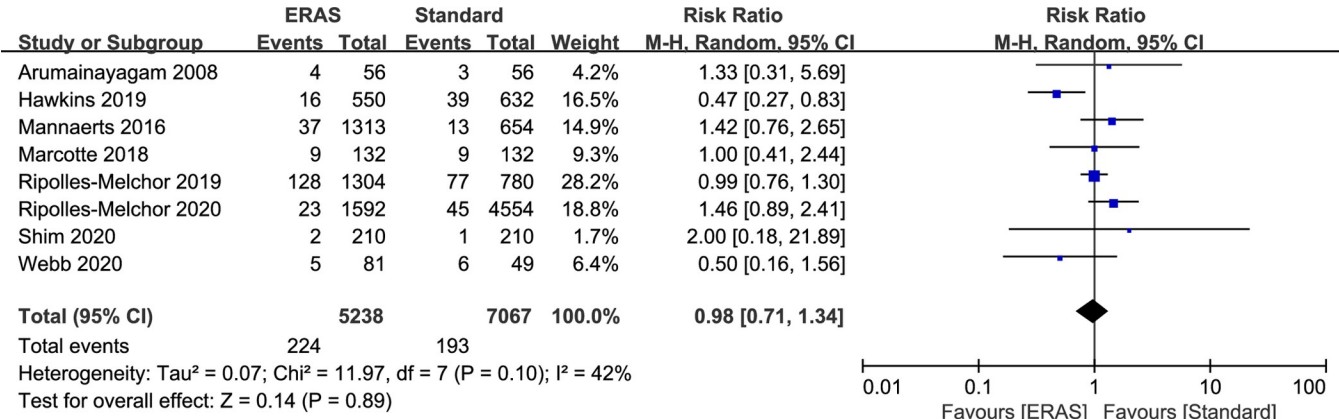

**Fig 7. Results of subgroup analysis results of 30-day reoperation rate in ERAS protocol versus standard care.** CI = confidence interval; ERAS = enhanced recovery after surgery; RR = risk ratio.

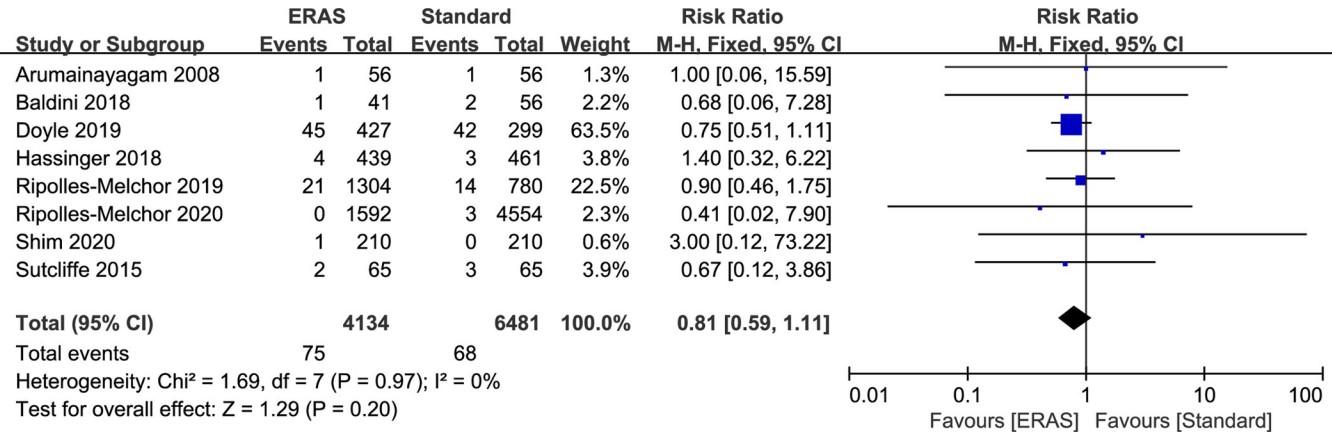

**Fig 8. Results of subgroup analysis results of mortality rate in ERAS protocol versus standard care.** CI = confidence interval; ERAS = enhanced recovery after surgery; RR = risk ratio.

minimize surgical stress, shorten LOS, reduce postoperative morbidity and reduce costs. ERAS has been widely used in abdominal and non-abdominal surgery [39, 40]. Nonetheless, the few relevant clinical trials and the unconvincing results are insufficient to support any firm conclusions about whether or to what extent ERAS protocols are associated with postoperative AKI. In contrast, the majority of studies in this area are retrospective or prospective cohort studies. This is the first meta-analysis evaluated the impact of enhanced recovery after surgery versus standard care on postoperative AKI. It is also one of the largest studies performed with regard to the number of studies and patients included. The studies we included were all high-quality cohort studies, most of our included studies were well-designed and having a low risk of bias; the accuracy of the results was verified by changing the effect model of the statistical method for sensitivity analysis by removing each study separately. Meta-analysis of this systematic review and comparative study showed that ERAS protocol did not increase the incidence of postoperative AKI but enabled faster postoperative recovery and shorter LOS without an increase in major postoperative complications, mortality, 30-day readmission, and reoperation rates. From the results, the ERAS protocol accelerated postoperative rehabilitation and reduced costs without increasing the incidence of postoperative AKI, and the safety regarding ERAS was further confirmed. A major caveat is that the evidence supporting the beneficial effects of ERAS protocols derives from a number of retrospective and prospective cohort studies.

AKI is very common and affects > 50% of patients in the ICU [41]. It is associated with increased mortality and high hospital costs. Traditional factors affecting patients often involve fluid replacement because of concerns about hypovolemia [42]. Although circulatory failure or hypoperfusion predisposes to AKI and timely fluid administration may be beneficial, there is growing evidence that excessive fluid resuscitation leads to adverse outcomes, including worsening renal function. In addition, there is clear evidence that certain fluids are nephrotoxic [43].

The pooled results from our meta-analysis indicated that different fluid replacement modalities had no significant effect on the incidence of postoperative AKI. The results of subgroup analysis showed that goal-directed fluid therapy and restrictive fluid management did not reduce the incidence of postoperative AKI in patients.

Patel [44] detailed physiological data confirmed that even short duration of intraoperative hypotension may contribute to perioperative AKI. Preventing postoperative AKI and ensuring

adequate renal perfusion are prerequisites. GDFT combined with vasoconstrictor drugs (prophylactic) can prevent the risk of organ hypoperfusion while avoiding the occurrence of postoperative tissue edema and cardiopulmonary complications caused by excessive volume supplementation. However, Patel's study failed to demonstrate a benefit of GDFT on postoperative increases in plasma creatinine. Nevertheless, to avoid postoperative renal injury, achieving preoperative oxygen supply seems to be essential [44]. Schmid's trial [45] showed that even without hemodynamic optimization, the overall care of patients undergoing high-risk surgery seems to be improved. The future role of GDFT may be to prevent volume depletion or overload. This can be improved by developing a volume therapy index, which is the next proposed goal in the field of critical care nephrology [46].

In ERAS protocols, it is generally recommended to avoid excessive fluid replacement [47]. Some small trials support restrictive fluid protocols [48, 49]. However, inappropriate fluid balance methods can be harmful. An international trial in the New England Journal of Medicine [50] compared a fluid restriction protocol with a liberal fluid replacement protocol. After 1 year, there was no significant improvement in survival but a significant increase in the risk of AKI for restrictive fluid therapy compared with liberal fluid therapy. These findings do not represent support for excessive fluid replacement. Instead, they showed that a protocol of appropriate liberal fluid replacement is safer than restrictive fluid management, which is similar to the results of our analysis.

The pooled results showed no heterogeneity among the three diagnostic modalities and no statistically significant difference from the pooled results, indicating that AKI diagnostic criteria are not the source of heterogeneity, and the different diagnostic modalities for AKI had no significant effect on the incidence of postoperative AKI in the ERAS protocol. In addition, some scholars have previously shown that the KDIGO diagnostic criteria are more liberal and accurate than the NSQIP, RFIFE, and AKIN criteria.

The first definition of AKI, called the RIFLE classification, was proposed in 2004 [36] and the AKIN classification, known as 'modified RIFLE', was in 2007 [51]. Recently, RIFLE and AKIN were merged into the KDIGO classification to provide a more liberal and easy-to-synthesize standard for clinical work and research [35]. In KDIGO, AKI was defined as an increase in serum creatinine (SCr) $\geq$ 0.3 mg / dl within 48 hours, or to $\geq$ 1.5 times the baseline value within 7 days, or urine volume $<$ 0.5 ml / kg / h in 6 hours [52]. The KDIGO criteria state that a number of preventive measures are implemented for high risk patients with AKI: including discontinuation and avoidance of nephrotoxic drugs, monitoring of SCr and urine output, hemodynamic monitoring of maintenance volume and perfusion pressure, use of substitutes for contrast agents, and maintenance of euglycemia [35]. Recently, a RCT study proved that the implementation of the biomarker guided KDIGO criteria significantly reduced the incidence of postoperative AKI after cardiac surgery compared with conventional treatment [53]. In addition, another study on patients undergoing major abdominal surgery also showed that the incidence of moderate to severe AKI in patients could be reduced and the length of ICU and hospital stay could be shortened by preusing a series of evaluation criteria [54].

The analysis results of other secondary outcome measures showed that patients with ERAS protocol had shorter postoperative hospital stay, and there was no statistical difference in 30-day readmission rate, reoperation rate and mortality compared with standard care, indicating that patients with ERAS had faster postoperative recovery and better. A large number of previous studies have shown that ERAS regimen reduces the incidence of common postoperative complications, accelerates the early recovery of intestinal function, accelerates the early postoperative mobilization of patients and reduces the cost of hospitalization [6, 55–58]. Obviously, ERAS protocol has many advantages compared with the standard care, and according to our study ERAS protocol has no significant impact on the incidence of postoperative AKI,

affirming the safety of ERAS protocol on renal function, which is conducive to the further development of ERAS protocol.

Our study has several limitations. First, there was moderate heterogeneity in the results of the meta-analysis of the primary outcome measures in this study. We performed a subgroup analysis for heterogeneity and found that different types of surgery had some impacts on the heterogeneity, and the sensitivity analysis results were also robust. We considered that because the results of the included cohort studies were different, some studies demonstrated that ERAS protocol did not affect the incidence of postoperative AKI, and others had opposite results, which may be the source of heterogeneity. Secondly, among the secondary outcome measures in this study, there was asymmetry in the funnel plot of LOS and the result of Egger regression test was P < 0.05, which may have publication bias; however, we found that the result was not significantly changed using the trim and fill analysis, and the result obtained by further using Begg adjusted rank correlation was negative. We inferred that the heterogeneity was the source of asymmetry in funnel plot and the result of egger regression test was P < 0.05. Moreover, all of the studies we included were cohort studies because the implementation of ERAS has become more and more popular in recent years. Most of the studies on ERAS compared with standard protocol were retrospective cohort studies and there were few high-quality RCT studies, so more RCT studies need to be included in the future to further prove the safety of ERAS protocol on postoperative AKI. Finally, the positive effect of NSAIDs on postoperative AKI has been well-known by the majority of physicians [58–60]. Since NSAIDs were used in all ERAS protocols included in our study, we could not verify whether NSAIDs in ERAS protocols have a decisive role in the incidence of postoperative AKI or whether they have a preventive effect. More studies are needed in the future to compare the proportion of NSAIDs, fluid management, hemodynamic management and other factors in impacting the incidence of postoperative AKI.

## Conclusions

In conclusion, this meta-analysis suggests that ERAS protocols do not increase readmission or reoperation rates and mortality, while significantly reducing LOS and effectively achieving rapid recovery. More importantly, the ERAS protocol was shown to have no promoting effect on the incidence of postoperative AKI. Even if GDFT and restrictive fluid management cannot avoid the occurrence of postoperative AKI, the ERAS protocol is still worth recommending and its safety is further confirmed. Further confirmation of the relationship between risk factors associated with postoperative AKI in ERAS protocols through randomized controlled trials is needed in the future.

## Supporting information

**S1 Fig.** A. Funnel plots for primary outcome (the rate of postoperative AKI); B. Sensitivity analysis plots for primary outcome. Abbreviations: AKI, acute kidney injury.
(DOCX)

**S2 Fig.** A. Funnel plots for LOS; B. Sensitivity analysis plots for LOS. Abbreviations: LOS = length of stay.
(DOCX)

**S3 Fig.** A. Funnel plots for 30-day readmission rate; B. Sensitivity analysis plots for 30-day readmission rate.
(DOCX)

**S4 Fig.** A. Funnel plots for 30-day reoperation rate; B. Sensitivity analysis plots for 30-day reoperation rate.
(DOCX)

**S5 Fig.** A. Funnel plots for mortality rate; B. Sensitivity analysis plots for mortality rate.
(DOCX)

**S1 Appendix. PUBMED search equation.**
(DOCX)

**S1 Checklist. PRISMA checklist.**
(DOC)

**S1 File.**
(RM5)

**S1 Raw material.**
(DOCX)

## Author Contributions

**Data curation:** Whenzhen Shen, Zehao Wu, Yunlu Wang.

**Formal analysis:** Whenzhen Shen, Zehao Wu, Yunlu Wang, Yi Sun, Anshi Wu.

**Investigation:** Zehao Wu, Yi Sun.

**Methodology:** Anshi Wu.

**Software:** Zehao Wu, Yunlu Wang, Yi Sun.

**Supervision:** Anshi Wu.

**Validation:** Zehao Wu.

**Visualization:** Anshi Wu.

**Writing – original draft:** Whenzhen Shen, Zehao Wu.

**Writing – review & editing:** Anshi Wu.

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
