## [Decision Letter · Decision Letter 0]

23 Feb 2021

PONE-D-20-39089

Impact of enhanced recovery after surgery (ERAS) protocol versus standard of care on postoperative acute kidney injury (AKI): A meta-analysis

PLOS ONE

Dear Dr.

   Anshi Wu 

Thank you for submitting your manuscript to PLOS ONE. After careful consideration, we feel that it has merit but does not fully meet PLOS ONE’s publication criteria as it currently stands. Therefore, we invite you to submit a revised version of the manuscript that addresses the points raised during the review process.

I would appreciate if pay a careful attention in your response to the reviewers' comments.

We look forward to receiving your revised manuscript.

Kind regards,

Ehab Farag, MD FRCA FASA

Academic Editor

PLOS ONE

Journal Requirements:

Reviewers' comments:

Reviewer's Responses to Questions

**Comments to the Author**

1. Is the manuscript technically sound, and do the data support the conclusions?

Reviewer #1: Yes

Reviewer #2: Yes

2. Has the statistical analysis been performed appropriately and rigorously? 

Reviewer #1: Yes

Reviewer #2: Yes

3. Have the authors made all data underlying the findings in their manuscript fully available?

Reviewer #1: Yes

Reviewer #2: Yes

4. Is the manuscript presented in an intelligible fashion and written in standard English?

Reviewer #1: Yes

Reviewer #2: No

5. Review Comments to the Author

Reviewer #1: Thank you for inviting me to review the article “Impact of enhanced recovery after surgery (ERAS) protocol versus standard of care on postoperative acute kidney injury (AKI): A meta-analysis”. Overall it is a solid study, whose results may impact our clinical practice. Below are several points need clarification.

L103 Q1: the part of statistical analysis is too general and short. More specific writing will be appreciated.

1) “With random effect model” was mentioned in the abstract but not in the statistical method part. Which models were used to do analysis? Please clarify. Only after I read the figure 3, I was able to assume your model would be Random-effects models with inverse variance weighting.

2) Please clarify “if the literature only gives odds ratio (OR) or hazard ratio (HR), and the required data cannot be calculated, it is approximate to RR value.” In the figure 3, RR was estimated from random effect model but not from reported OR/HR. Since most of included studies are retrospective cohort studies (biggest limitation), adjusted OR or HRs were reported, it may be a different direction with estimated RR (<1 vs. >1), indicating different conclusions. Please mention current RR results from random-effect model if same as original OR/HR (i.e., if different directions (RR< 1 and original OR > 1) or significant level p value change from >0.05 to <0.05). Please clarify.

3) At least for primary outcome AKI, please clarify how many sensitivity analyses were performed and for what (or why). It is same for subgroup analyses. I did not get a whole picture how the analyses were planned until I read your results later. For example, L148 “by removing each study individually and changing the effect model”.

Q2 For figure 3/4, since subgroup and overall analysis were together, the column weighting is calculated for overall but not in subgroups. Please explain in the legend or consider to add one more column for subgroup weighting.

Q3 Sample size justification and Post-hoc power analyses since most of included studies are retrospective cohort studies?

Reviewer #2: Overall, the authors performed nice meta-analysis on the association between ERAS and post-op AKI. The analysis is comprehensive, and the results are detailed. Below are some minor suggestions.

1. Inclusion criteria (Ln 83-85) indicated that cohort studies were included, while line 119 and121 indicated trials. There are also multiple places in the results where trials were mentioned. Please check and clarify.

2. The method section should be more detailed on the statistical methods used. Line 111 “effect model” is not a stand-alone statistical term – I assume you want to say fixed or random effect models.

3. In the result for AKI incidence, please also include the overall incidence of AKI (case number and incidence rate) of all the studies included.

4. The result mentioned that baseline characteristics differed widely studies (ln 125-126). I would suggest including basic patient characteristics (e.g. age and BMI) in the summary table. It will be interesting to see if cohort characteristics will be a source of heterogeneity, even through adjusted results are used for the meta-analysis.

5. Please also comment on whether the surgery types are comparable to each other to be included in the same meta-analysis (e.g., colorectal vs Total hip and knee arthroplasty).

6. The authors should proof-read the manuscript and check for gramma errors. Especially, (1) methods and results should be in past tense instead of future tense; (2) avoid long and clustered sentences – the logic can be hard to follow.

6. PLOS authors have the option to publish the peer review history of their article (what does this mean?). If published, this will include your full peer review and any attached files.

Reviewer #1: **Yes: **Mi Wang

Reviewer #2: No

---

## [Author Response · Author response to Decision Letter 0]

13 Apr 2021

April 8, 2021

Dear editors and reviewers,

On behalf of my colleagues, I am herewith submitting the revised manuscript titled “Impact of enhanced recovery after surgery (ERAS) protocol versus standard of care on postoperative acute kidney injury (AKI): A meta-analysis” with manuscript number PONE-D-20-39089 for consideration of publication in PLOS ONE. We would like to thank the editors and reviewers’ work devoted to our manuscript and we are very grateful for their valuable suggestions. We have fully addressed each concern and hope that this revised manuscript is now acceptable.

Revisions are indicated within the text using highlighted changes (in red), as requested in the journal’s guidelines.

Please see below for our detailed responses, which are highlighted in bold blue fonts.

Reviewer #1: Thank you for inviting me to review the article “Impact of enhanced recovery after surgery (ERAS) protocol versus standard of care on postoperative acute kidney injury (AKI): A meta-analysis”. Overall, it is a solid study whose results may impact our clinical practice. Below are several points need clarification.

Response: We thank the reviewer for the favorable analysis of our original submission and for highlighting the significance of our studies.

L103 Q1: the part of statistical analysis is too general and short. More specific writing will be appreciated.

Response: Thank you for your advice. Our statistical description is indeed too general, and after joint research and discussion by all authors, the statistical method part has been supplemented in detail and meticulously modified, as page 3, line 105-115.

1) “With random effect model” was mentioned in the abstract but not in the statistical method part. Which models were used to do analysis? Please clarify. Only after I read the figure 3, I was able to assume your model would be Random-effects models with inverse variance weighting.

Response: Thank you for your questions. We deeply apologize for the inappropriate presentation of the effect model in the statistical approach, for which we redescribed the choice of the effect model as follows: The Q and I2 statistics were used to determine heterogeneity, I2 was defined as: 100%×(Q – df)/Q, where Q was Cochrane’s heterogeneity statistic, df was the degree of freedom, I2 < 50% indicated that there was no significant heterogeneity, a fixed-effect model was used, and conversely > 50% used a random-effect model. The overall effect was assessed by Z test using a random effects model (Inverse Variance method) and statistical significance was determined when the 95% CIs did not include the value of 1.0 for the RR or 0 for the MD. See page 3 line 105 and 109 for manuscript revision.

2) Please clarify “if the literature only gives odds ratio (OR) or hazard ratio (HR), and the required data cannot be calculated, it is approximate to RR value.” In the figure 3, RR was estimated from random effect model but not from reported OR/HR. Since most of included studies are retrospective cohort studies (biggest limitation), adjusted OR or HRs were reported, it may be a different direction with estimated RR (<1 vs. >1), indicating different conclusions. Please mention current RR results from random-effect model if same as original OR/HR (i.e., if different directions (RR< 1 and original OR > 1) or significant level p value change from >0.05 to <0.05). Please clarify.

Response: Thanks for your correction. We deeply apologize for the use error of the estimation method of OR/RR values here, for which we have removed the incorrect part of the description of the methodological statistical analysis part in the text and redescribed the calculation of RR values as follows: The overall effect was assessed by Z test using a random effects model (Inverse Variance method) and statistical significance was determined when the 95% CIs did not include the value of 1.0 for the RR or 0 for the MD. See page 3 line 105 for manuscript revision.

3) At least for primary outcome AKI, please clarify how many sensitivity analyses were performed and for what (or why). It is same for subgroup analyses. I did not get a whole picture how the analyses were planned until I read your results later. For example, L148 “by removing each study individually and changing the effect model”.

Response: We deeply agree with your concern. In order to correct the possible bias and make the combined results as robust as possible, we found the sources that may affect the stability of the outcome according to the previous studies, such as the type of surgery and the type of study. The sensitivity analysis was used to eliminate the relevant studies one by one, and it was found that the combined results did not change significantly, indicating that such factors had little effect on the results, and the combined analysis results were robust. In addition, there was moderate heterogeneity according to the combined AKI results of the primary outcome, and we strived to find the source of heterogeneity according to the stratified analysis of possible influencing factors. Subgroup analyses were performed on the fluid management mode, as well as AKI diagnostic criteria. This has been revised, see page 3, line 110.

Q2 For figure 3/4, since subgroup and overall analysis were together, the column weighting is calculated for overall but not in subgroups. Please explain in the legend or consider to add one more column for subgroup weighting.

Response: Thank you for your questions. Because figure 3 and 4 combine the overall and subgroup analysis, we have only calculated the column weighting of the overall and not the subgroup analysis. In this regard, we added a column of weighted proportion for different subcomponent layers in figure 3/4. The modified figure may be statistically more complete and persuasive. The modified figure has been re-uploaded and covered the previous figure. See Figure 3/4 (revision) or the following figure for details.

Figure 3 (revision):

Figure 4 (revision):

Q3 Sample size justification and Post-hoc power analyses since most of included studies are retrospective cohort studies?

Response: Thank you for your advice. We agree with your proposal as most of the included studies are retrospective cohort studies, statistical power for the primary outcome should be calculated using G power software, and Post hoc: Compute power-given achieved α, sample size, and effect size should be selected as the type of power analysis. Because the sample size of this study is sufficient, the results suggest that the statistical power is high (power = 0.9999764), so the possibility that the significant results reported in this study truly reflect the real effect is high. The specific software operation process is as shown in the figure below. See page 3, line 116 in the Method part and page 4, line 154 in the Results part for the detailed modification.

Reviewer #2: Overall, the authors performed nice meta-analysis on the association between ERAS and post-op AKI. The analysis is comprehensive, and the results are detailed. Below are some minor suggestions.

Response: We thank the reviewer for the favorable analysis of our original submission and for highlighting the significance of our studies.

Comment 1:

Inclusion criteria (Ln 83-85) indicated that cohort studies were included, while line 119 and121 indicated trials. There are also multiple places in the results where trials were mentioned. Please check and clarify.

Response: Thanks for your correction. We deeply apologize for the wrong wording here, and indeed the cohort study and trial cannot be confused, and we have carefully reviewed the full text and revised the full text where all relevant wording is not appropriate, please see page 3, line 123.

Comment 2:

The method section should be more detailed on the statistical methods used. Line 111 “effect model” is not a stand-alone statistical term – I assume you want to say fixed or random effect models.

Response: Thank you for your questions. Our statistical description is indeed too general, and after joint research and discussion by all authors, the statistical method part has been supplemented in detail and meticulously modified. We also deeply apologize for the inappropriate presentation of the effect model in the statistical approach, for which we redescribed the choice of the effect model as follows: The Q and I2 statistics were used to determine heterogeneity, I2 was defined as: 100%×(Q – df)/Q, where Q was Cochrane’s heterogeneity statistic, df was the degree of freedom, I2 < 50% indicated that there was no significant heterogeneity, a fixed-effect model was used, and conversely > 50% used a random-effect model. The overall effect was assessed by Z test using a random effects model (Inverse Variance method) and statistical significance was determined when the 95% CIs did not include the value of 1.0 for the RR or 0 for the MD. See page 3 line 105 and 115 for manuscript revision.

Comment 3:

In the result for AKI incidence, please also include the overall incidence of AKI (case number and incidence rate) of all the studies included.

Response: Thanks for your suggestion! We traced the original data of all patients in 19 cohort studies and found that 831 of all 17,205 patients were confirmed to have AKI after operation, with an overall incidence rate of about 4.83%, including 461 patients in ERAS regimen and 370 patients in standard treatment who had AKI after operation. This has been revised, please see page 3, line 128.

Comment 4:

The result mentioned that baseline characteristics differed widely studies (ln 125-126). I would suggest including basic patient characteristics (e.g. age and BMI) in the summary table. It will be interesting to see if cohort characteristics will be a source of heterogeneity, even through adjusted results are used for the meta-analysis.

Response: Thanks for your advice! We deeply agree with you that, even if we use multiple methods to correct confounding factors and bias, such as funnel plot, sensitivity analysis and Egger’s regression test, it is very important to observe whether the baseline characteristics of the two cohorts are heterogeneous and the source of heterogeneity. For this reason, we added the inclusion of new baseline indicators in the baseline indicator table, including age and BMI. Unfortunately, some individual studies did not report the detailed data of the above baseline indicators. See Table 1 for detailed modifications, which page 8, line 426.

Comment 5:

Please also comment on whether the surgery types are comparable to each other to be included in the same meta-analysis (e.g., colorectal vs Total hip and knee arthroplasty).

Response: We deeply agree with your concern. We have considered your doubts and discussed and analyzed them before, but it is our fault to not explain them in this paper, and we are very sorry for this. Studies have shown that the incidence of AKI is higher after cardiac surgery, which may be related to inflammatory factors and ischemia, but the pooled results remain unchanged after the use of sensitivity analysis in the cardiac surgery cohort included in this study, indicating that the cardiac surgery cohort included in this study has little effect on the reliability and robustness of the primary outcome results. While the remaining included cohort studies were all non-cardiac surgeries, as previous studies have shown that ERAS protocol has been widely used in major abdominal surgeries, orthopedic surgeries, sensitivity analysis and funnel plot have shown robust results, so the cohorts included in this study were comparable in terms of the occurrence of postoperative AKI between different surgery types.

Comment 6:

The authors should proof-read the manuscript and check for gramma errors. Especially, (1) methods and results should be in past tense instead of future tense; (2) avoid long and clustered sentences – the logic can be hard to follow.

Response: Thanks for your suggestion! We apologize for the improper wording that has affected your reading. We have carefully reviewed the full text and proofread it again, revising the grammatical problems and tense errors, streamlining some long and redundant sentences, strengthening the logical framework, making sentences and paragraphs smoother and increasing readability.

We thank you and reviewer for the in depth reading of the manuscript. We believe the changes have significantly increased the impact of the manuscript.

Sincerely,

Zehao Wu, MD.

Corresponding author: Anshi Wu, MD, PhD.

Department of Anesthesiology

Beijing Chaoyang Hospital, Capital Medical University, Beijing, China

Phone number: +86-135-1101-0883. Email address: wuanshi88cy@163.com

---

## [Editor Report · Decision Letter 1]

28 Apr 2021

Impact of enhanced recovery after surgery (ERAS) protocol versus standard of care on postoperative acute kidney injury (AKI): A meta-analysis

PONE-D-20-39089R1

Dear Dr.

   Anshi Wu 

We’re pleased to inform you that your manuscript has been judged scientifically suitable for publication and will be formally accepted for publication once it meets all outstanding technical requirements.

Kind regards,

Ehab Farag, MD FRCA FASA

Academic Editor

PLOS ONE
---

## [Editor Report · Acceptance letter]

10 May 2021

PONE-D-20-39089R1 

Impact of enhanced recovery after surgery (ERAS) protocol versus standard of care on postoperative acute kidney injury (AKI): A meta-analysis 

Dear Dr. Wu:

I'm pleased to inform you that your manuscript has been deemed suitable for publication in PLOS ONE. Congratulations! Your manuscript is now with our production department. 

Kind regards, 

on behalf of

Dr. Ehab Farag 

Academic Editor

PLOS ONE